# Quantum Secret Aggregation Utilizing a Network of Agents

**Michael Ampatzis** [†] [ID] **and Theodore Andronikos** *,[†] [ID]

Department of Informatics, Ionian University, 7 Tsirigoti Square, 49100 Corfu, Greece
* Correspondence: andronikos@ionio.gr
† These authors contributed equally to this work.

**Abstract:** Suppose that the renowned spymaster Alice controls a network of spies who all happen to be deployed in different geographical locations. Let us further assume that all spies have managed to get their hands on a small, albeit incomplete by itself, secret, which actually is just a part of a bigger secret. In this work, we consider the following problem: given the above situation, is it possible for the spies to securely transmit all these partial secrets to the spymaster so that they can be combined together in order to reveal the big secret to Alice? We call this problem, which, to the best of our knowledge, is a novel one for the relevant literature, the quantum secret aggregation problem. We propose a protocol, in the form of a quantum game, that addresses this problem in complete generality. Our protocol relies on the use of maximally entangled GHZ tuples, shared among Alice and all her spies. It is the power of entanglement that makes possible the secure transmission of the small partial secrets from the agents to the spymaster. As an additional bonus, entanglement guarantees the security of the protocol, by making it statistically improbable for the notorious eavesdropper Eve to steal the big secret.

**Keywords:** quantum entanglement; GHZ states; quantum cryptography; quantum secret sharing; quantum secret aggregation





## 1. Introduction

The rapidly growing dependence and continuous development of many prominent network-based technologies, such as the internet of things and cloud-based computing, have resulted in an ever-growing need for more reliable and robust security protocols that can protect our present infrastructure from malicious individuals or parties. Our current security protocols, which base their security upon a set of computationally difficult mathematical problems, such as the factorization problem, have been proven reliable, at least for the time being. Nonetheless, they have also been proven vulnerable against more sophisticated attacks that incorporate the use of quantum algorithms and quantum computers. Despite the fact that most of these quantum algorithms were theoretically developed a couple of decades ago, such as the two famous algorithms developed by Peter Shor and Lov Grover [1,2], for many years, there was no immediate threat of such attacks. That was simply because the technology was not mature enough to even produce a quantum computer capable of surpassing the 100 qubit barrier, let alone of having the qubit capacity required to actually break these encryption protocols.

Today, with the monumental breakthrough of IBM's new quantum computers, which managed to surpass the 100 qubit barrier [3] last year and was then immediately followed a year later by their most recent 433 qubit quantum processor named Osprey [4] that managed to quadruple the previous processor's qubit capacity, the landscape has changed dramatically. It is now clear that we are much closer to successfully developing a viable fully working quantum computer than we originally anticipated. Thus, the need has arisen for an immediate upgrade of our security protocols before they become a critical liability to our communication infrastructure. This inherent vulnerability of the current protocols has led to a plethora of initiatives from various countries and organizations,

all aiming at establishing new and novel approaches for solving the ever-more critical problem of secure communication [5]. Among the various attempts to provide a robust solution for this problem, two new scientific fields emerged, namely, the field of post-quantum or quantum-resistant cryptography and the field of quantum cryptography. Despite the confusing similarities in their names, these fields attempt to solve the problem by implementing radically different strategies. Specifically, the field of quantum-resistant cryptography is trying to maintain the philosophy of the previous era by still relying on the use of mathematical problems, albeit of a more complex nature, such as supersingular elliptic curve isogenies and supersingular isogeny graphs, solving systems of multivariate equations, and lattice-based cryptography. On the other hand, the field of quantum cryptography is trying to establish security by relying on the fundamental principles of quantum mechanics, such as the monogamy of entangled particles, the no-cloning theorem and nonlocality.

Presently, the most prominent of the two aforementioned fields is that of post-quantum cryptography [6–9]. This can be attributed to the fact that the successful implementation of such protocols does not require any changes in the current infrastructure. However, in our view, this is just a temporary state of affairs, caused by the inherent limitations of the current technology. The long-term future of cryptography can be nothing other than quantum cryptography, which is a crucial contemporary research topic. This is due to the overwhelming advantages of the fundamental properties of quantum mechanics, which not only allow us to protect our information, but also efficiently transmit information using entangled states, as first proposed by Arthur Ekert [10]. In his E91 quantum key distribution protocol (QKD for short), Ekert proved that key distribution is possible with the use of EPR pairs. After this landmark discovery by Ekert, the field of quantum cryptography witnessed rapid growth in the development of entanglement-based QKD protocols [11–16]. This has solidified the technique's importance and has prompted the research community to expand the field by experimenting with other cryptographic primitives, such as quantum secret sharing.

The cryptographic primitive of secret sharing or secret splitting in its more elementary form can be described as a game between two spatially separated groups. The first group typically consists of one player who wants to share a secret message with the other group. The latter consists of the rest of the players who will receive the secret message split into multiple pieces. By itself, each piece does not contain any valuable information, but if all the players in the second group were to combine their pieces, the secret message would be revealed. Understandably, one may regard this cryptographic primitive as nothing more than a scaled-up key distribution protocol, designed in order to accommodate more than two people. However, this would be an unfair assessment that overlooks the step of dividing the secret message into multiple pieces. This step offers a crucial advantage by providing security against malicious individuals that have managed to infiltrate the second group with the goal of covertly acquiring the secret message by forcing every player, honest or dishonest, to participate in the process that unlocks the secret message (see the recent [17] for more details).

Secret sharing schemes are vital for providing security to new and emerging technologies, such as cloud computing, cloud storage [18,19] or blockchain [20]. These technologies require multiple parties to communicate with each other, accommodating the possibility that some of them might be malicious users, who want to take advantage of the system. Therefore, the research on quantum secret sharing has come a long way from the simple proof of concept by Hillery et al. [21], and Cleve et al. [22], who pioneered this field. All this progress has led to numerous research proposals and schemes that are continuously expanding the field to this day [23–28]. At the same time, multiple experimental demonstrations involving real-world scenarios have been attempted by the researchers in [29–32], and even schemes for non-binary quantum secret-sharing protocols that rely on the use of qudits instead of qubits [33–36] have been proposed.

This work tackles a problem that could be considered the inverse of the standard quantum secret-sharing scheme. Specifically, we consider a setting where there is a network of agents who are all distributed in different locations. The spies have explicit orders to operate only on a need-to-know basis, meaning they must avoid any form of communication among themselves, and only report directly to the spymaster Alice, who resides in a different location from all her agents. Moreover, we assume that all the spies have managed to get their hands on a small, albeit incomplete by itself, secret, which actually is only a part of a bigger secret. A critical parameter in this situation is that all of the partial secrets must be combined together if one is to reveal the big picture. The ultimate goal of this scenario is to successfully transmit all the partial secrets gathered by the agents to the spymaster. Furthermore, caution is required during the transmission phase in order to safeguard against a possible breach of confidentiality from any unknown adversary. Thus, going quantum seems the way to go. We refer to this problem as quantum secret aggregation, and we give a protocol that solves this task in the form of a game. The use of games should not diminish the seriousness or importance of the problem, but we hope to make its presentation more entertaining and memorable. Certainly, this is not the first time games, such as coin tossing, etc., have been used in quantum cryptography (see [37] and recently [16,17]). Quantum games have captured the interest of many researchers since their inception in 1999 [38,39]. In many situations, quantum strategies seem to outperform classical ones [40–42]. This holds not only for iconic classical games, such as the prisoners' dilemma [39,43], but also for abstract quantum games [44]. As a matter of fact, there is no inherent restriction on the type of a classical system that can be transformed to a quantum analogue, as even political institutions may be amenable to this process [45].

One can easily envision some real-life applications where the ability to have such a single-step, coherent, efficient, and fast multi-party communication protocol, with the additional specification that every partial secret is required in order to unveil the bigger secret, is beneficial or even crucial. We give the following two examples as a proof of concept.

**Example 1** (The Treasure Map). *This example serves as a simple metaphor for elaborate real-life situations, where digitized visual, acoustic, or similar types of information, must be combined in order to reveal the complete picture. Note that each piece of information by itself is incomplete. The proposed protocol can produce the whole picture as fast as possible.*

*Imagine that Bob and Charlie, two of Alice's agents, have each managed to uncover a part of a map containing precise instructions for finding a precious treasure. The previous owner had torn the map into two separate pieces that were kept in different locations as a security precaution. Each partial map gives incomplete instructions, and only by putting them together can the treasure be found. This state of affairs is visually depicted in Figure 1. Bob and Charlie must transmit their corresponding partial maps to Alice, so that she may claim the treasure.*

*It goes without saying that this example can be readily generalized to an arbitrary number of involved agents.*

**Example 2** (Clandestine Voting). *This example should also be seen as a metaphor for special instances of voting procedures.*

*Consider a situation where the members of an organization, who happen to be spatially separated, play the role of Alice's agents. These agents are called to vote (secretly) for an important decision, e.g., the next chairman of the board. Alice assumes the role of the trusted referee that guarantees the honest conclusion of the voting process. The agents place their votes that correspond to their individual secrets, and which must be transmitted to Alice who is responsible for counting the votes. The extra step where every vote must be combined in order to unlock the winner (final outcome) provides us with the assurance that a malicious or corrupt individual will not be able to intercept the votes during the transmission phase or change them at the collection facility before the outcome is determined.*

*A possible variation of the above situation would be the case where the shareholders of an international corporation are called to vote for the election of the new board of directors or for hiring*

*high-level staff. Each shareholder may be responsible for the selection of a specific position (or positions). We may view Alice as corresponding with the election committee and her agents with the shareholders. All votes (selections) must be combined together so as to fill all the empty positions of the corporation.*

### BOB's secret

Find the Island of
Get to the misty river
Walk up the brown hill
Dig two meters left of

### CHARLIE's secret

the buried treasure.
in the east of the green forest.
and turn left.
the big oak tree.

**Figure 1.** This figure depicts Bob and Charlie's incomplete maps, that must be sent to Alice, so that she may dig-up the treasure.

**Contribution**. This paper poses and solves a novel, to the best of our knowledge, problem in the general context of quantum cryptographic protocols. We refer to it as the quantum secret aggregation problem because it involves aggregating many small secrets in order to reveal a bigger secret. The underlying setting visualizes a completely distributed network of agents, each in possession of a small secret. These secrets contain incomplete information by themselves, and only by combining them together can the bigger secret be revealed. Therefore, the agents have to send their partial secrets to the spatially separated Alice, which is our famous spymaster. The operation must be completed in the most secure way possible, as there are eavesdroppers eager to intercept their communications and steal the big secret. To address this problem, we present the quantum secret aggregation protocol as a game. The solution outlined is completely general, as the number of players can be scaled arbitrarily as needed and all $n$ players are assumed to reside in different positions in space. Obviously, the solution still holds, even if a subset of the players are located in the same place. Security is enforced because of the integral role of entanglement in the protocol. The use of maximally entangled GHZ tuples shared among Alice and all her spies not only makes possible the secure transmission of the small partial secrets from the agents to Alice, but also guarantees the security of the protocol by making it statistically improbable for the notorious eavesdropper Eve to obtain the big secret.

**Organization**. The structure of this paper is the following. Section 1 provides an introduction to the subject along with some relevant references. Section 2 is a brief exposition on GHZ states and the phenomenon of entanglement. Section 3 rigorously defines the problem at hand, while Section 4 explains in detail the quantum secret aggregation protocol. Section 5 presents a small example of the protocol executed using Qiskit. Section 6 is devoted to the security analysis on a number of possible attacks from Eve, and, finally, Section 7 contains a summary and a brief discussion on some of the finer points of this protocol.

## 2. A Brief Reminder about GHZ States

Nowadays, most quantum protocols designed to securely transmit keys, secrets, and information in general rely on the power of entanglement. Entanglement is a hallmark property of the quantum world. As this phenomenon is absent from the everyday world, it is considered counterintuitive by some. However, from the point of view of quantum cryptography and quantum computation, this strange behavior is seen as a precious resource, which is the key to achieve quantum teleportation and unconditionally secure information transmission.

Thus, it comes as no surprise that this work also utilizes quantum entanglement in a critical manner in order to implement the proposed protocol of quantum secret aggregation. Specifically, our protocol relies on the maximally entangled $n$-tuples of qubits, i.e., qubits that are in the literature referred to as the $|GHZ_n\rangle$ state. Present-day quantum computers

can produce arbitrary GHZ states using various quantum circuits. A methodology for constructing efficient such circuits is given in [46]. The resulting quantum circuits are efficient in the sense that they require $\lg n$ steps to generate the $|GHZ_n\rangle$ state. One such circuit that generates the $|GHZ_5\rangle$ state using the IBM Quantum Composer [47] is shown in Figure 2. The dotted lines are a helpful visualization that allows us to distinguish "time slices" within which the CNOT gates are applied in parallel. Figure 3, which is also from the IBM Quantum Composer, indicates the state vector description of the $|GHZ_5\rangle$ state.

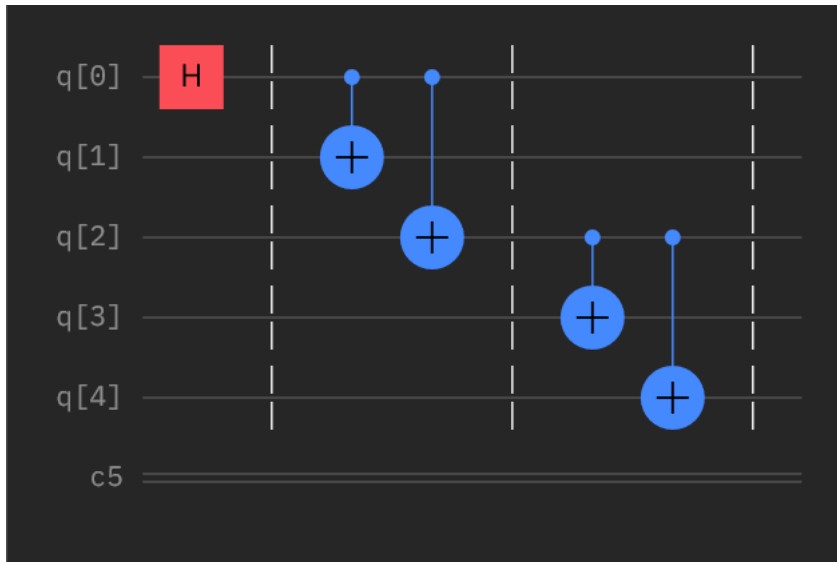

**Figure 2.** The above (efficient) quantum circuit in Qiskit can entangle 5 qubits in the $|GHZ_5\rangle = \frac{|0\rangle|0\rangle|0\rangle|0\rangle|0\rangle+|1\rangle|1\rangle|1\rangle|1\rangle|1\rangle}{\sqrt{2}}$ state. Following the same pattern, we can construct efficient quantum circuits that entangle $n$ qubits in the $|GHZ_n\rangle$ state.

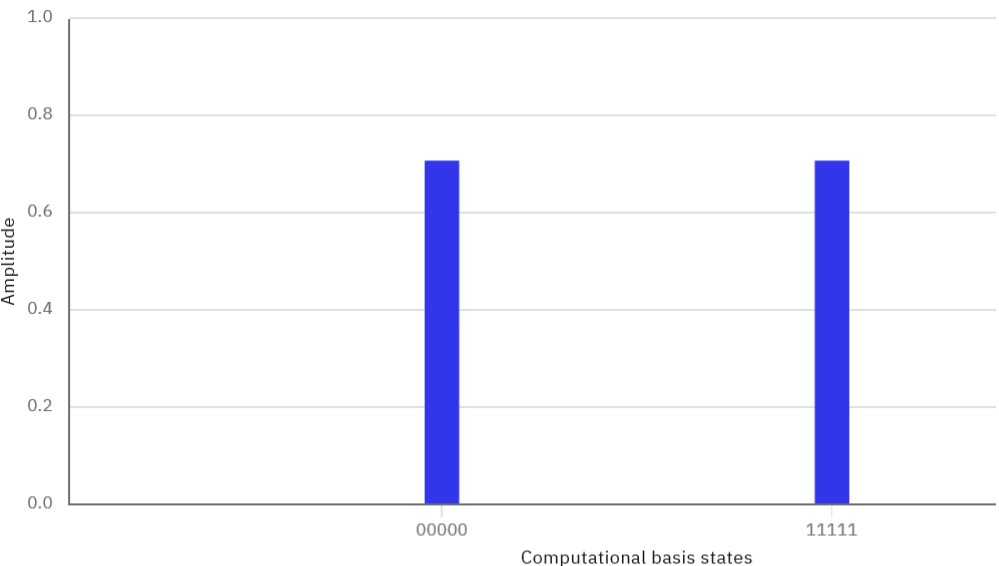

**Figure 3.** This figure depicts the state vector description of 5 qubits that are entangled in the $|GHZ_5\rangle$ state.

Let us assume that we are given a composite quantum system made up of $n$ individual subsystems, where each subsystem contains just a single qubit. As explained above, it is possible to entangle all these $n$ of the composite system qubits in the $|GHZ_n\rangle$ state. In such a case, the mathematical description of the state of the composite system is the following:

$$|GHZ_n\rangle = \frac{1}{\sqrt{2}} \left( |0\rangle_{n-1} |0\rangle_{n-2} \dots |0\rangle_0 + |1\rangle_{n-1} |1\rangle_{n-2} \dots |1\rangle_0 \right), \tag{1}$$

where the subscript $i$, $0 \le i \le n-1$, is used to indicate the qubit belonging to subsystem $i$.

It is expedient and necessary to generalize the above setting so that each individual subsystem is a quantum register and not just a single qubit. In this more general situation, each of the $n$ subsystems is a quantum register $r_i$, where $0 \le i \le n-1$, which has $m$ qubits. The characteristic property of this setting is that the corresponding qubits of all the $n$ registers are entangled in the $|GHZ_n\rangle$ state. This means that all the $n$ qubits in position $j$, $0 \le j \le m-1$, of the registers $r_0, r_1 \dots, r_{n-1}$ are entangled in the $|GHZ_n\rangle$ state. Figure 4 provides a visual depiction of this situation, where the corresponding qubits comprising the $|GHZ_n\rangle$ $n$-tuple are drawn with the same color. In this composite system, there exist $m$, i.e., the number of qubits in each register, $|GHZ_n\rangle$ $n$-tuples. Thus, the global state of the composite system is captured by the $m$-fold tensor product $|GHZ_n\rangle^{\otimes m}$, and its mathematical description is

$$|GHZ_n\rangle^{\otimes m} = \frac{1}{\sqrt{2^m}} \sum_{\mathbf{x} \in \{0,1\}^m} |\mathbf{x}\rangle_{n-1} \dots |\mathbf{x}\rangle_0 , \tag{2}$$

where $\mathbf{x} \in \{0,1\}^m$ ranges through all the $2^m$ basis kets.

Equation (2) can be proved by an easy induction on $m$. For $m = 1$, Equation (2) reduces to (1), and trivially holds. Let us assume that, according to the induction hypothesis, (2) holds for $m$. We shall prove that (2) also holds for $m+1$. Indeed, by invoking (1) and (2), the computation shown below completes the proof by induction.

$$
\begin{aligned}
|GHZ_n\rangle^{\otimes m+1} &= |GHZ_n\rangle^{\otimes m} \otimes |GHZ_n\rangle \\
&= \frac{1}{\sqrt{2^m}} \sum_{\mathbf{x} \in \{0,1\}^m} |\mathbf{x}\rangle_{n-1} \dots |\mathbf{x}\rangle_0 \otimes \frac{1}{\sqrt{2}} \left( |0\rangle_{n-1} |0\rangle_{n-2} \dots |0\rangle_0 + |1\rangle_{n-1} |1\rangle_{n-2} \dots |1\rangle_0 \right) \\
&= \frac{1}{\sqrt{2^{m+1}}} \sum_{\mathbf{x} \in \{0,1\}^m} |\mathbf{x}0\rangle_{n-1} \dots |\mathbf{x}0\rangle_0 + |\mathbf{x}1\rangle_{n-1} \dots |\mathbf{x}1\rangle_0 \\
&= \frac{1}{\sqrt{2^{m+1}}} \sum_{\mathbf{x} \in \{0,1\}^{m+1}} |\mathbf{x}\rangle_{n-1} \dots |\mathbf{x}\rangle_0 .
\end{aligned}
\tag{3}
$$

A composite system consisting of $n$ quantum registers $r_0, \ldots, r_{n-1}$. Each register has $m$ qubits and the corresponding qubits are entangled in the $|GHZ_n\rangle$ state, i.e., the $n$ qubits in the $j^{th}$ position, $0 \le j \le m-1$, of the registers $r_0, \ldots, r_{n-1}$ constitute an $n$-tuple entangled in the $|GHZ_n\rangle$ state.

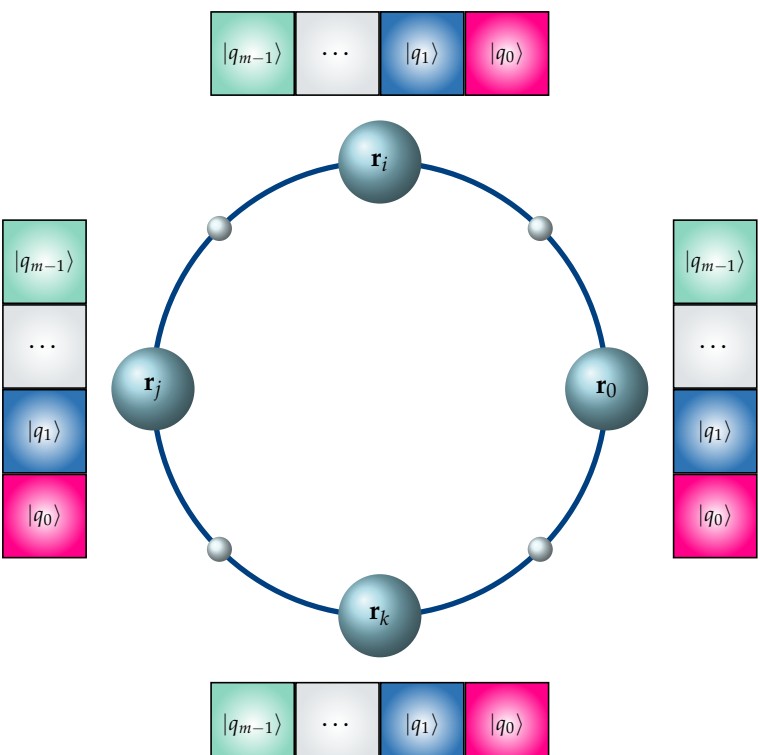

**Figure 4.** This figure visualizes the situation where each of the $n$ subsystems is a quantum register $r_i$, $0 \le i \le n-1$, that has $m$ qubits, and the corresponding qubits in all the registers are entangled in the $|GHZ_n\rangle$ state. This means that qubit $|q_0\rangle$ of register $r_0$, qubit $|q_0\rangle$ of register $r_1, \ldots,$ qubit $|q_0\rangle$ of register $r_{n-1}$ constitute an $n$-tuple entangled in the $|GHZ_n\rangle$ state. As a visual confirmation of this fact, these qubits have been drawn with the same color. The same holds for all $n$ qubits in position $j$, $1 \le j \le n-1$, of registers $r_0, \ldots, r_{n-1}$ and the coloring scheme employed aims to emphasize this fact.

### 3. The Problem of Quantum Secret Aggregation

In the current section, we rigorously define the problem of Quantum Secret Aggregation, simply referred to as QSA from now on. To the best of our knowledge, this is the first time that this problem is posed and solved in the relevant literature. Informally, QSA can be considered the inverse of quantum secret sharing (QSS for short). The latter focuses on how a single entity (usually called Alice) can securely transmit a secret to a group of two or more agents. Typically in QSS, Alice is in a different location from her agents; however, the agents are assumed to be in the same location, which implies that they can readily exchange information. In contrast, in QSA, we assume that Alice and her agents are all in different locations, and this time it is the agents that want to securely transmit a part of the secret to Alice. Each agent has only a small part of the secret, and no two agents possess secrets with common fragments. Alice requires all the parts in order to decipher the secret.

**Definition 1** (Quantum Secret Aggregation). *Let us assume that the following hold.*

*(A₁)*　　*There are $n-1$ spatially separated agents $Agent_0, \ldots, Agent_{n-2}$. The number of agents is totally arbitrary, i.e., it may be odd or even. Each agent possesses of a* partial secret key $\mathbf{p}_i$, $0 \le i \le n-2$.

*(A₂)*　　*Every partial secret key is unique and is known only to the corresponding agent. Furthermore, there is no information redundancy among the partial secret keys, i.e., no one can be inferred from the rest.*

*(A₃)*　　*The partial secret keys are, in general of* different length. *This means that, denoting by $|\mathbf{p}_i|$ the length of $\mathbf{p}_i$, in general, it holds that $|\mathbf{p}_i| \ne |\mathbf{p}_j|$, $0 \le i \ne j \le n-2$.*

*(A₄)*　　*The agents want to securely send their secret key to the spymaster Alice, who is also in an entirely different location.*

*(A₅)*　　*Alice wants to discover the* complete secret key, *denoted by $\mathbf{s}$. This can only be done by combining all the partial secret keys $\mathbf{p}_0, \ldots, \mathbf{p}_{n-2}$.*

*(A₆)*　　*The length of the complete secret key, denoted by m, is the sum of the lengths of all the partial secret keys: $m = |\mathbf{p}_0| + \cdots + |\mathbf{p}_{n-2}|$. The agents send the length of their partial key to Alice, thus enabling her to compute m. Subsequently, Alice sends m to all her spies so that it becomes common knowledge to Alice and her agents.*

*(A₇)*　　*The whole operation must be executed with utmost secrecy, due to the presence of the eavesdropper Eve.*

*The* quantum secret aggregation *problem asks how to establish a protocol that will guarantee that Alice and her agents achieve their goal.*

In view of the fact that $Agent_i$ possesses the partial key $\mathbf{p}_i$, $0 \le i \le n-2$, we can make the following observations.

- Implicit in the definition of the problem is the assumption that Alice has assigned a specific ordering to her ring of agents and all her agents are aware of this ordering. This simply means that not only Alice but also all agents know who are $Agent_0, \ldots, Agent_{n-2}$.

- Definition 1 explicitly allows the partial secret keys to be of different length, which is far more probable and realistic.

- Although neither Alice nor her agents know the partial secret keys (except their own), they all know their lengths $|\mathbf{p}_0|, \ldots, |\mathbf{p}_{n-2}|$. This does not compromise the secrecy factor because knowing the length of a secret key does not reveal its contents.

From an algorithmic perspective, it is convenient to have a standard length for all partial secret keys. This prompts the following definition.

**Definition 2** (Extended Partial Secret Key). *Each $Agent_i$, $0 \le i \le n-2$, constructs from her partial secret key $\mathbf{p}_i$ her* extended *partial secret key $\mathbf{s}_i$, which is defined as*

$$\mathbf{s}_i = \underbrace{0 \cdots 0}_{k \text{ times}} \, \mathbf{p}_i \, \underbrace{0 \cdots 0}_{l \text{ times}}, \tag{4}$$

*where $k = |\mathbf{p}_{n-2}| + \cdots + |\mathbf{p}_{i+1}|$ and $l = |\mathbf{p}_{i-1}| + \cdots + |\mathbf{p}_0|$.*

This simple construction enforces uniformity among the agents since they all end up having extended keys of length *m*, even though their partial keys will in general be of different lengths, and greatly simplifies the construction of the quantum circuit. In practice, in order to achieve uniformity, the agents must initially disclose the lengths of their partial secret keys $|\mathbf{p}_i|$ to Alice for her to calculate **m** and advise her agents on how to construct their extended partial secret key $\mathbf{s}_i$, an action which can be done through a public channel. Additionally, it enables us to derive the next simple and elegant formula connecting the complete secret key **s** with the extended partial secret keys $\mathbf{s}_0, \ldots, \mathbf{s}_{n-2}$:

$$\mathbf{s} = \mathbf{s}_0 \oplus \cdots \oplus \mathbf{s}_{n-2} . \tag{5}$$

## 4. The Quantum Secret Aggregation Protocol

We now present the proposed QSA protocol as a game, aptly named the QSA game. In this game, there are $n$, $n \geq 3$, players, which can be conceptually divided into two groups. Alice alone makes the first group, which is the recipient of the secret information from distant sources. These sources are the $n - 1$ agents in the spy ring that constitute the second group. The proposed protocol is general enough to accommodate an arbitrary number of agents. To thoroughly describe the QSA game, we carefully distinguish the phases in its progression.

### 4.1. Initialization Phase through the Quantum Channel

This game utilizes entanglement. As a matter of fact, its successful completion relies on the use of entanglement. So, it is necessary, before the main part of the protocol commences, to create the required number, which is denoted by $m$, of $n$-tuples of qubits entangled in the $|GHZ_n\rangle$ state. Such entangled tuples can be produced by a contemporary quantum computer, for instance, using a quantum circuit such as the one shown in Figure 2. These $|GHZ_n\rangle$ tuples can be produced by Alice or by another trusted source, which can even be a satellite [48]. Figure 5 depicts the former situation. We note, however, that our protocol does not depend on which source actually creates the entangled tuples. The crucial requirement is that they are produced and sent through the quantum channel so that they may populate the input registers of Alice and all her agents.

> **THE QUANTUM CHANNEL**
> Alice sends through the quantum channel $m$ qubits to each of the $n - 1$ spatially distributed agents in her spy network. Each qubit is entangled in the $|GHZ_n\rangle$ state.

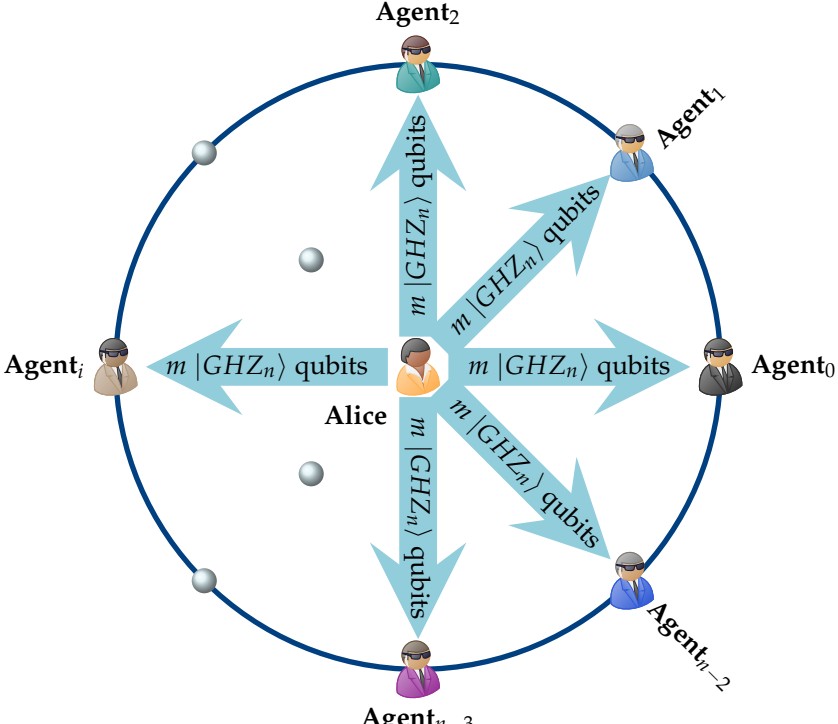

**Figure 5.** The above figure depicts the situation where Alice herself initiates the protocol by creating and sending through the quantum channel to each of the $n - 1$ spatially distributed agents in her spy network $m$ qubits, each one of them entangled in the $|GHZ_n\rangle$ state.

### 4.2. Input Phase in the Local Quantum Circuits

The purpose of the QSA game from Alice's point of view is to aggregate all the partial secret keys $\mathbf{p}_0, \ldots, \mathbf{p}_{n-2}$ from her $n-1$ agents, in order to reveal the complete secret key $\mathbf{s}$. All the $n-1$ partial keys are absolutely necessary for this, as they are distinct and nonoverlapping, i.e., there is no information redundancy among them. From the perspective of the individual agents, the operation is strictly on a need-to-know basis, which means that after the completion of the protocol, they gain no additional information that they did not know already.

The QSA protocol successfully accomplishes this feat by employing the quantum circuit shown in Figure 6. There, we show the individual quantum circuits employed by Alice and her $n-1$ agents Agent$_0$, ..., Agent$_{n-2}$. Table 1 explains the abbreviations that are used in the quantum circuit depicted in Figure 6. It is important to emphasize that this is a distributed quantum circuit made up of $n$ individual, spatially separated and private circuits. It is the phenomenon of entanglement that strongly correlates the individual subcircuits, forming, in effect, a composite distributed circuit. The state vectors $|\psi_0\rangle$, $|\psi_1\rangle$, $|\psi_2\rangle$, $|\psi_3\rangle$ and $|\psi_4\rangle$ describe the evolution of the composite system. The $n$ individual subcircuits have obvious similarities, and some important differences, as summarized in Table 2. Let us also clarify that for consistency, we follow the Qiskit [49] convention in the ordering of qubits by placing the least significant at the top and the most significant at the bottom.

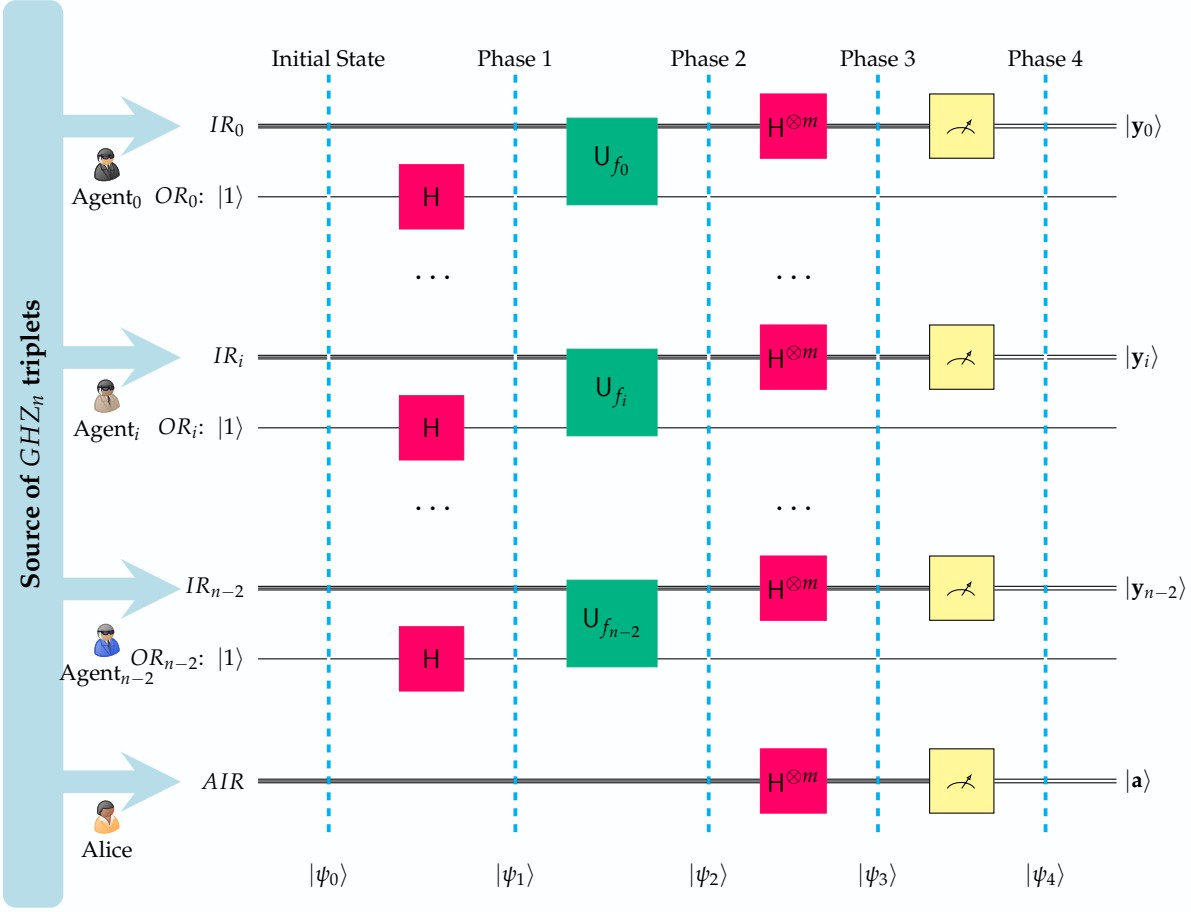

**Figure 6.** The above figure shows the quantum circuits employed by Alice and her agents. We point out that these circuits are spatially separated, but, due to entanglement, strongly correlated forming a composite system. The state vectors $|\psi_0\rangle$, $|\psi_1\rangle$, $|\psi_2\rangle$, $|\psi_3\rangle$ and $|\psi_4\rangle$ describe the evolution of the composite system.

In our subsequent analytical mathematical description of the QSA game, we use the typical convention of writing the contents of quantum registers in boldface, e.g., $|\mathbf{x}\rangle = |x_{m-1}\rangle \ldots |x_0\rangle$, for some $m \geq 1$. Moreover, apart from Equation (2), we will make use of the two other well-known formulas given below (see any standard textbook, such as [50] or [51]).

$$H|1\rangle = \frac{1}{\sqrt{2}}(|0\rangle - |1\rangle) = |-\rangle \tag{6}$$

$$H^{\otimes m}|\mathbf{x}\rangle = \frac{1}{\sqrt{2^m}} \sum_{\mathbf{z} \in \{0,1\}^m} (-1)^{\mathbf{z} \cdot \mathbf{x}} |\mathbf{z}\rangle , \tag{7}$$

where $|\mathbf{z}\rangle = |z_{m-1}\rangle \ldots |z_0\rangle$ and $\mathbf{z} \cdot \mathbf{x}$ is the inner product modulo 2, defined as

$$\mathbf{z} \cdot \mathbf{x} = z_{m-1}x_{m-1} \oplus \cdots \oplus z_0 x_0 . \tag{8}$$

**Table 1.** This table contains the notations and abbreviations that are used in Figure 6.

| Notations and Abbreviations | |
| :---: | :---: |
| **Symbolism** | **Explanation** |
| $n$ | Number of players (Alice plus her $n-1$ agents) |
| $m$ | Length of the secret key **s**, equal to the number of qubits in the Input Registers of Alice & every one of her agents |
| AIR | Alice's $m$-qubit Input Register |
| IR$_i$ | The $m$-qubit Input Register of Agent$_i$, $0 \leq i \leq n-2$ |
| OR$_i$ | The single-qubit Output Register of Agent$_i$, $0 \leq i \leq n-2$ |

**Table 2.** Differences and similarities among the $n$ subcircuits depicted in Figure 6.

| Differences and Similarities | |
| :---: | :---: |
| **Differences** | **Similarities** |
| Alice's circuit lacks Output Register | All circuits contain an $m$-qubit Input Register |
| Alice does not apply any function | All agents' circuits contain an Output Register |
| Every agent applies a different function $f_i$ | All Output Registers are initialized to $|1\rangle$ |
| | All circuits apply the $m$-fold Hadamard transform on their Input Register prior to measurement |

The circuit of Figure 6 contains $n$ input registers, all having $m$ qubits: one for Alice and one for each of her agents. The qubits in the $j^{th}$, $0 \leq j \leq m-1$, position of the input registers form an $n$-tuple entangled in the $|GHZ_n\rangle$ state. Additionally, each agent, but not Alice, is in possession of a single qubit output register.

Based on Equation (2), the initial state $|\psi_0\rangle$ of the circuit shown in Figure 6 can be written as

$$|\psi_0\rangle = \frac{1}{\sqrt{2^m}} \sum_{\mathbf{x} \in \{0,1\}^m} |\mathbf{x}\rangle_A |1\rangle_{n-2} |\mathbf{x}\rangle_{n-2} \ldots |1\rangle_0 |\mathbf{x}\rangle_0 . \tag{9}$$

In Equation (9), $|\mathbf{x}\rangle_A$ designates the contents of Alice's input register, $|1\rangle_i$, $0 \leq i \leq n-2$, is the state of the agents' output registers, and $|\mathbf{x}\rangle_i$, $0 \leq i \leq n-2$, denotes the contents of the input registers of the $n-1$ agents. In what follows, the subscripts $A$ and $0, 1, \ldots, n-2$ are utilized in an effort to distinguish between the local registers of Alice and Agent$_0$, $\ldots$, Agent$_{n-2}$, respectively.

The first phase of the protocol begins when all the agents apply the Hadamard transform to their respective output register, driving the system to the next state $|\psi_1\rangle$

$$|\psi_1\rangle = \frac{1}{\sqrt{2^m}} \sum_{\mathbf{x}\in\{0,1\}^m} |\mathbf{x}\rangle_A |-\rangle_{n-2} |\mathbf{x}\rangle_{n-2} \cdots |-\rangle_0 |\mathbf{x}\rangle_0 . \tag{10}$$

At this point, each of the $n-1$ agents transmits her secret. Since this is the most important part of the protocol, we explain in detail how this task is implemented. Agent$_i$, $0 \le i \le n-2$, defines a function that is based on her extended partial secret key $\mathbf{s}_i$, namely

$$f_i(\mathbf{x}) = \mathbf{s}_i \cdot \mathbf{x} , \ 0 \le i \le n-2 . \tag{11}$$

Agent$_i$, $0 \le i \le n-2$, uses function $f_i$ to construct the unitary transform $U_{f_i}$, which, as is typical of many quantum algorithms, acts on both output and input registers, producing the following output:

$$U_{f_i} : |y\rangle |\mathbf{x}\rangle \to |y \oplus f(\mathbf{x})\rangle |\mathbf{x}\rangle . \tag{12}$$

Taking into account (10), which asserts that for every agent the state of the output register is $|-\rangle$, and (11), Formula (12) becomes

$$U_{f_i} : |-\rangle |\mathbf{x}\rangle \to (-1)^{\mathbf{s}_i \cdot \mathbf{x}} |-\rangle |\mathbf{x}\rangle . \tag{13}$$

Hence, the cumulative action of the unitary transforms $U_{f_i}$, $0 \le i \le n-2$ sends the quantum circuit to the next state:

$$\begin{aligned}
|\psi_2\rangle &= \frac{1}{\sqrt{2^m}} \sum_{\mathbf{x}\in\{0,1\}^m} |\mathbf{x}\rangle_A (-1)^{\mathbf{s}_{n-2}\cdot\mathbf{x}} |-\rangle_{n-2} |\mathbf{x}\rangle_{n-2} \cdots (-1)^{\mathbf{s}_0\cdot\mathbf{x}} |-\rangle_0 |\mathbf{x}\rangle_0 \\
&= \frac{1}{\sqrt{2^m}} \sum_{\mathbf{x}\in\{0,1\}^m} (-1)^{(\mathbf{s}_{n-2}\oplus\cdots\oplus\mathbf{s}_0)\cdot\mathbf{x}} |\mathbf{x}\rangle_A |-\rangle_{n-2} |\mathbf{x}\rangle_{n-2} \cdots |-\rangle_0 |\mathbf{x}\rangle_0 \\
&\overset{(5)}{=} \frac{1}{\sqrt{2^m}} \sum_{\mathbf{x}\in\{0,1\}^m} (-1)^{\mathbf{s}\cdot\mathbf{x}} |\mathbf{x}\rangle_A |-\rangle_{n-2} |\mathbf{x}\rangle_{n-2} \cdots |-\rangle_0 |\mathbf{x}\rangle_0 .
\end{aligned} \tag{14}$$

At this point, the complete secret key is implicitly encoded in the state of the circuit. It remains to be deciphered by Alice, as explained in the next subsection.

*4.3. Retrieval Phase*

Subsequently, Alice and all her spies apply the $m$-fold Hadamard transformation to their input registers. The next state of the circuit is shown below. Please note that henceforth, and in order to make the remaining formulas more readable and understandable, we have chosen to omit the output registers; they have served their intended purpose and will no longer be of any use.

$$\begin{aligned}
|\psi_3\rangle &= \frac{1}{\sqrt{2^m}} \sum_{\mathbf{x}\in\{0,1\}^m} (-1)^{\mathbf{s}\cdot\mathbf{x}} H^{\otimes m} |\mathbf{x}\rangle_A \, H^{\otimes m} |\mathbf{x}\rangle_{n-2} \cdots H^{\otimes m} |\mathbf{x}\rangle_0 \\
&\overset{(7)}{=} \frac{1}{\sqrt{2^m}} \sum_{\mathbf{x}\in\{0,1\}^m} (-1)^{\mathbf{s}\cdot\mathbf{x}} \left( \frac{1}{\sqrt{2^m}} \sum_{\mathbf{a}\in\{0,1\}^m} (-1)^{\mathbf{a}\cdot\mathbf{x}} |\mathbf{a}\rangle_A \right) \\
&\quad \left( \frac{1}{\sqrt{2^m}} \sum_{\mathbf{y}_{n-2}\in\{0,1\}^m} (-1)^{\mathbf{y}_{n-2}\cdot\mathbf{x}} |\mathbf{y}_{n-2}\rangle_{n-2} \right) \cdots \left( \frac{1}{\sqrt{2^m}} \sum_{\mathbf{y}_0\in\{0,1\}^m} (-1)^{\mathbf{y}_0\cdot\mathbf{x}} |\mathbf{y}_0\rangle_0 \right) \\
&= \frac{1}{(\sqrt{2^m})^{n+1}} \sum_{\mathbf{x}\in\{0,1\}^m} \sum_{\mathbf{a}\in\{0,1\}^m} \sum_{\mathbf{y}_{n-2}\in\{0,1\}^m} \cdots \sum_{\mathbf{y}_0\in\{0,1\}^m} (-1)^{(\mathbf{s}\oplus\mathbf{a}\oplus\mathbf{y}_{n-2}\oplus\cdots\oplus\mathbf{y}_0)\cdot\mathbf{x}} |\mathbf{a}\rangle_A |\mathbf{y}_{n-2}\rangle_{n-2} \cdots |\mathbf{y}_0\rangle_0 .
\end{aligned} \tag{15}$$

The above formula looks complicated but it can be simplified by invoking an important property of the inner product modulo 2 operation. If $|\mathbf{c}\rangle = |c_{m-1}\rangle \ldots |c_0\rangle \neq |0\rangle^{\otimes m}$ is a fixed basis ket, then for precisely half of the basis kets $|\mathbf{x}\rangle$, $\mathbf{c} \cdot \mathbf{x}$ will be 0 and for the remaining half, $\mathbf{c} \cdot \mathbf{x}$ will be 1. In the special case, where $|\mathbf{c}\rangle = |0\rangle^{\otimes m}$, then for *every* basis ket $|\mathbf{x}\rangle$, $\mathbf{c} \cdot \mathbf{x} = 0$. Applying this property to Equation (15), we conclude that if

$$\mathbf{a} \oplus \mathbf{y}_{n-2} \oplus \cdots \oplus \mathbf{y}_0 = \mathbf{s} \, , \tag{16}$$

then, for each $\mathbf{x} \in \{0,1\}^m$, the expression $(-1)^{(\mathbf{s}\oplus\mathbf{a}\oplus\mathbf{y}_{n-2}\oplus\cdots\oplus\mathbf{y}_0)\cdot\mathbf{x}}$ becomes $(-1)^0 = 1$. Therefore, the sum $\sum_{\mathbf{x}\in\{0,1\}^m}(-1)^{(\mathbf{s}\oplus\mathbf{a}\oplus\mathbf{y}_{n-2}\oplus\cdots\oplus\mathbf{y}_0)\cdot\mathbf{x}}$ equals $2^m$. In contrast, when $\mathbf{a} \oplus \mathbf{y}_{n-2} \oplus \cdots \oplus \mathbf{y}_0 \neq \mathbf{s}$, the sum reduces to 0. This is typically written in a compact way as

$$\sum_{\mathbf{x}\in\{0,1\}^m}(-1)^{(\mathbf{s}\oplus\mathbf{a}\oplus\mathbf{y}_{n-2}\oplus\cdots\oplus\mathbf{y}_0)\cdot\mathbf{x}} = 2^m \delta_{\mathbf{s},\mathbf{a}\oplus\mathbf{y}_{n-2}\oplus\cdots\oplus\mathbf{y}_0} \, . \tag{17}$$

In view of (17), we may express state $|\psi_3\rangle$ more succinctly as

$$|\psi_3\rangle = \frac{1}{(\sqrt{2^m})^{n-1}} \sum_{\mathbf{a}\in\{0,1\}^m} \sum_{\mathbf{y}_{n-2}\in\{0,1\}^m} \cdots \sum_{\mathbf{y}_0\in\{0,1\}^m} |\mathbf{a}\rangle_A \, |\mathbf{y}_{n-2}\rangle_{n-2} \cdots |\mathbf{y}_0\rangle_0 \, . \tag{18}$$

The fundamental property of the QSA protocol, as encoded in Equations (17) and (18) states that the contents of the input registers of Alice and all her $n-1$ agents can not vary completely freely and independently. The presence of tuples entangled in the $|GHZ_n\rangle$ state during the initialization of the quantum circuit has manifested itself in state $|\psi_3\rangle$ in what we call the **fundamental correlation property**. This property asserts that in *each term* of the linear combination described by $|\psi_3\rangle$, the states $|\mathbf{a}\rangle_A$, $|\mathbf{y}_{n-2}\rangle_{n-2}, \ldots, |\mathbf{y}_0\rangle_0$ of the $n$ players' input registers are correlated by the following constraint:

$$\mathbf{a} \oplus \mathbf{y}_{n-2} \oplus \cdots \oplus \mathbf{y}_0 = \mathbf{s} \, . \tag{19}$$

The quantum part of the QSA protocol is completed when all players, i.e., Alice and her secret agents $\text{Agent}_0, \ldots, \text{Agent}_{n-2}$ measure their input registers, which results in the final state $|\psi_4\rangle$ of the quantum circuit.

$$|\psi_4\rangle = |\mathbf{a}\rangle_A \, |\mathbf{y}_{n-2}\rangle_{n-2} \cdots |\mathbf{y}_0\rangle_0 \, , \quad \text{for some} \quad \mathbf{a}, \mathbf{y}_0, \ldots, \mathbf{y}_{n-2} \in \{0,1\}^m \, , \tag{20}$$

where $\mathbf{a}, \mathbf{y}_0, \ldots, \mathbf{y}_{n-2}$ are correlated via (19). The unique advantage of entanglement has led to this situation: although the contents of each of the $n$ input registers may deceptively seem completely random to each player, in fact they are not. The distributed quantum circuit of Figure 6, considered a composite system, ensures that the final contents of the input registers satisfy the fundamental correlation property, as expressed by (19).

One final step remains. $\text{Agent}_0, \ldots, \text{Agent}_{n-2}$ must all send the contents of their input registers $\mathbf{y}_0, \ldots, \mathbf{y}_{n-2}$, respectively, to Alice, so as to allow Alice to uncover the big secret $\mathbf{s}$. This can be achieved by communicating through the classical channel. Figure 7 gives a mnemonic visualization of the conclusion of the QSA protocol.

The use of a public channel by the agents to broadcast their measurements will not compromise the security of the protocol for two reasons. First, the transmitted information $\mathbf{y}_i$, $0 \leq i \leq n-2$, is completely unrelated to the extended partial secret $\mathbf{s}_i$. The latter cannot be recovered from the former. Secondly, in the general case, even if Eve combines all the measurements $\mathbf{y}_0, \ldots, \mathbf{y}_{n-2}$, she still needs $\mathbf{a}$ in order to discover the secret message $\mathbf{s}$. There is, of course, the special case where $\mathbf{a} = \mathbf{0}$. In such a case, Eve has all the information she needs to find the secret message $\mathbf{s}$, although she might not know it, i.e., she might have no way to know that Alice's measurement is $\mathbf{0}$. Thus, to secure our protocol from this eventuality, we dictate that Alice should request the repetition of the whole process in the event that the contents of her input registers are all zero after the final measurement.

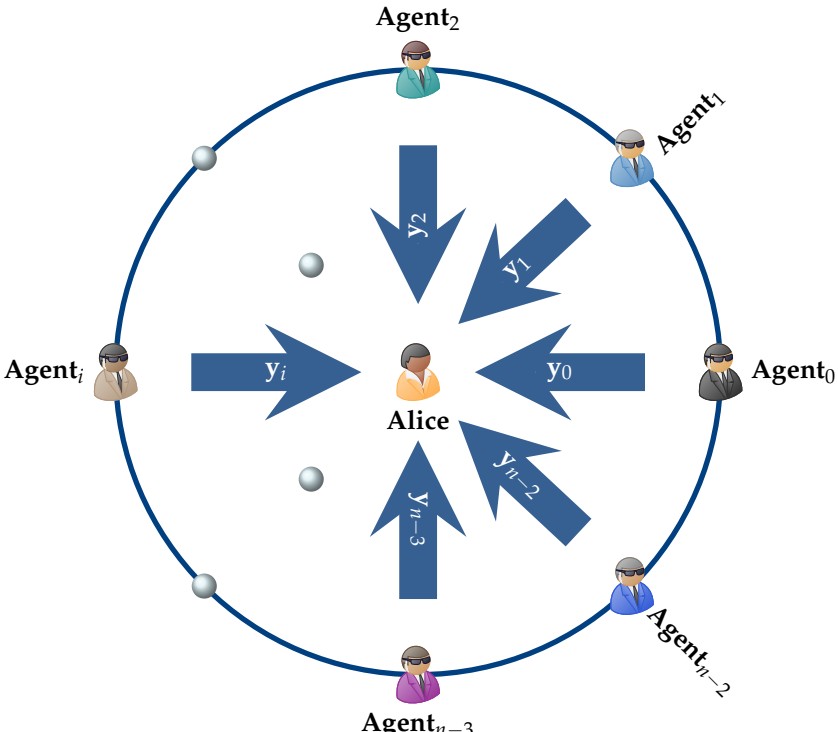

**Figure 7.** The above figure visualizes the conclusion of the QSA protocol when the $n-1$ spatially distributed agents in the spy network send to Alice through the classical channel the final measurements $\mathbf{y}_0, \ldots, \mathbf{y}_{n-2}$ of their input registers.

## 5. A Toy Scale Example Demonstrating the QSA Protocol

In this section, we present a toy scale example that should be viewed as a proof of concept about the viability of the QSA protocol. The resulting quantum circuit is illustrated in Figure 8. It was designed and simulated using IBM's *Qiskit* open source SDK ([49]) and, in particular, the Aer provider utilizing the high performance *qasm* simulator for simulating quantum circuits [52]. The measurements, of which only a small portion is shown in Figure 9, as their sheer number makes their complete visualization inexpedient, along with their corresponding probabilities were obtained by running the qasm simulator for 4096 shots.

In the current example, Alice's network consists of just two agents, none other than Bob and Charlie. All of them are in different locations. Bob's partial secret key is $\mathbf{p}_B = 10$ and Charlie's partial secret key is $\mathbf{p}_C = 01$. Hence, their extended partial secret keys are $\mathbf{s}_B = 1000$ and $\mathbf{s}_C = 0001$, and the complete secret key that ALice must uncover is $\mathbf{s} = 1001$. As we clarified above, the local quantum circuit of Figure 8 is best considered to be a proof of concept. This is because, at present, we are unable simulate in Qiskit the fact that Alice, Bob, and Charlie are spatially separated. An actual implementation of the QSA protocol would result in a distributed quantum circuit and not a local one as shown in Figure 8. Furthermore, we are also unable to directly specify a trusted third party source that generates the entangled GHZ triples, although Qiskit provides the ability to initialize the quantum circuit in specific initial state. In any case, we opted for the circuit itself to create the GHZ triples. Hence, these assumptions cannot be accurately reflected in the quantum circuit of Figure 8, and this example should be considered a faithful representation of a real-life scenario.

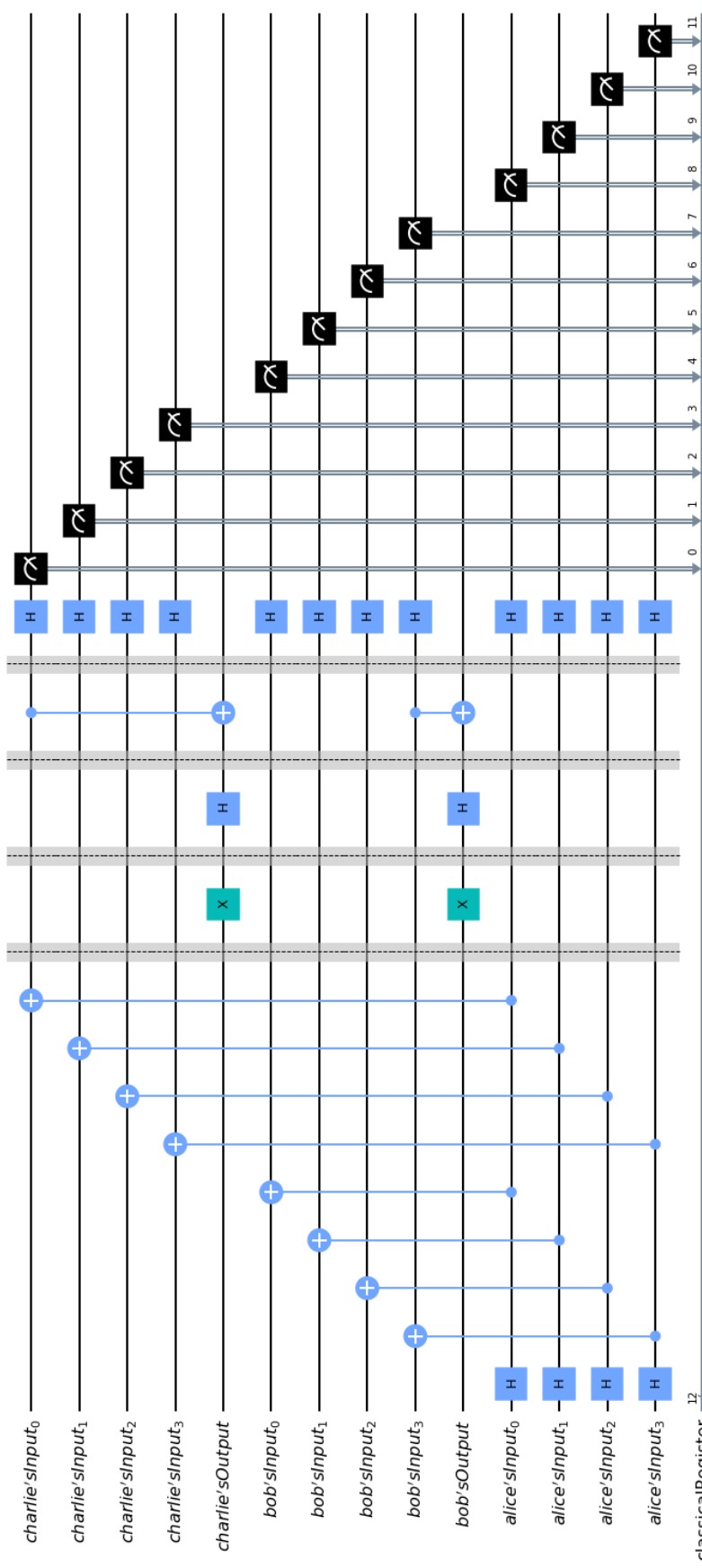

**Figure 8.** A toy scale quantum circuit simulating the QSA protocol, as applied to the spymaster Alice and her two agents Bob and Charlie.

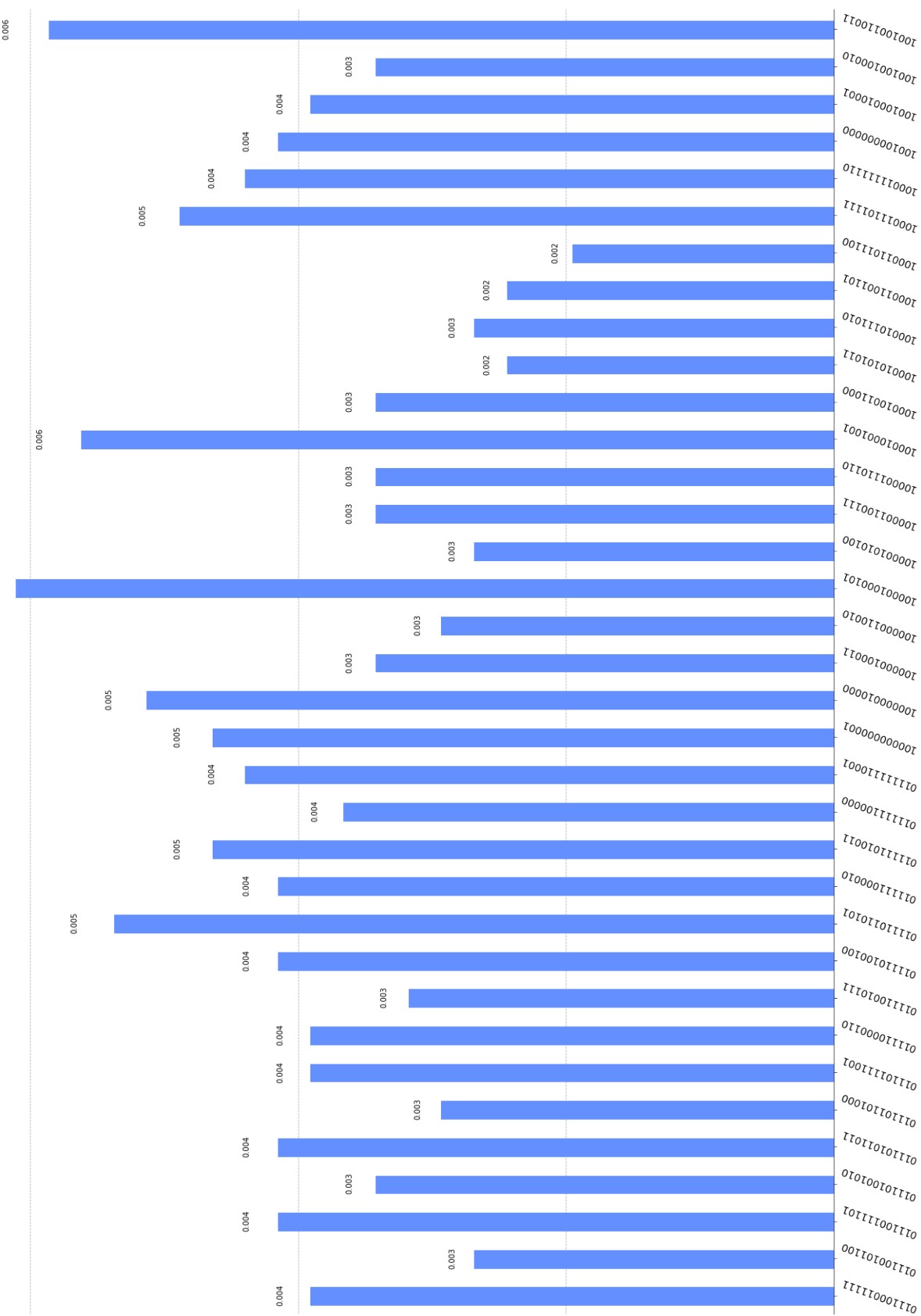

**Figure 9.** Some of the possible measurements and their corresponding probabilities for the circuit of Figure 8.

With all the above observations duly noted, we may verify that this simulation is indeed a localized version of the blueprint for the QSA protocol, as shown in Figure 6. The final measurements by Alice, Bob and Charlie will produce one of the $2^8 = 256$ equiprobable outcomes. Showing all these outcomes would result in an unintelligible figure so we opted for depicting only some of them in Figure 9. This figure also shows the corresponding probabilities for each outcome; it should not come as a surprise that they are not shown to be equiprobable as the theory expects since the figure resulted from a simulation run for 4096 shots. The important thing though is that every possible outcome satisfies the fundamental correlation property and verifies Equation (19). Therefore, ignoring the unlikely case that Alice measures $\mathbf{a} = 0000$ in her input register, Bob and Charlie, after measuring their input registers and obtaining $\mathbf{y}_B$ and $\mathbf{y}_C$, respectively, only have to send their measurements to Alice so that she can uncover the secret key.

## 6. Security Analysis of the QSA Protocol

### 6.1. Assumptions

In this section, we shall focus on analyzing several different attack strategies that a malicious individual, namely Eve, can incorporate against our protocol, with the goal of acquiring a piece of the secret message, or in the worst-case scenario, the complete message. This will allow us to establish the security of our protocol and its viability in practical applications. However, before we start with our analysis, it is crucial to first clarify two fundamental assumptions that we take for granted and serve as the basis of our security claims.

We begin by stating the first and most basic assumption, namely that quantum theory is correct and that we can use quantum mechanics to make accurate predictions about measurement outcomes. The reasoning behind this assumption is quite obvious due to the fact that if the underlying theory was false in one way or another, certain features of quantum mechanics, such as the no-cloning theorem [53], the monogamy of entanglement [54] or nonlocality [55], which are vital for any quantum cryptographic protocol, would not apply and thus, it would have been impossible to create a secure protocol.

The second assumption that we adopt is that quantum theory is complete and there are no other special properties or phenomena of quantum mechanics that we do not know. This means that Eve's movements are restricted by the laws of physics, and she cannot go beyond what is possible with quantum mechanics in order to acquire more information from her targets. This assumption by its very nature is not perfect, as the question regarding the completeness of quantum mechanics is still unresolved. However, the combination of the correctness of quantum mechanics, along with the requirement that free randomness exists implies that any future extension of quantum theory will not improve the predictive abilities of any player [56].

### 6.2. Intercept and Resend Attack

We start our security analysis by inspecting the first attack strategy, which of course is the most basic and intuitive type of an individual attack, known as intercept and resend or (I&R) attack. The main idea of this strategy is for Eve to get a hold of each photon coming from Alice or whoever is responsible for the distribution of the GHZ tuples to the rest of the players at the beginning of the protocol. Afterwards, Eve proceeds to measure them on some predefined basis and based on the result, to prepare a new photon and send it to the intended recipient. For this attack, it is rather obvious that in any of the aforementioned possible scenarios in which our protocol can be used, the GHZ tuples during the distribution phase of the protocol do not carry any information as regards the nature of the secret message. Thus, our SQA protocol is secure against this attack strategy.

### 6.3. PNS Attack

The next attack strategy, known as the photon number splitting attack (PNS), was first introduced by Huttner et al. [57] and further discussed and analyzed by Lütkenhaus

and Brassard et al. in [58,59]. Today, it is considered one of the most effective attack strategies that Eve can use against any protocol. This is because it exploits the fact that our current detectors are not 100% efficient and our photon sources do not emit single-photon signals all the time, meaning that there is a possibility for a photon source to produce multiple identical photons instead of only one. Therefore, in a realistic scenario, Eve can intercept these pulses coming from the player or the source responsible for the distribution of the GHZ tuples, take one photon from the multi-photon pulse and send the remaining photon(s) to their legitimate recipient undisturbed. In this scenario, Eve once again will not be able to acquire any information regarding the secret message or the random binary strings that will be used to unlock the secret key. This can be explained from the inherent nature of the QSA protocol, which leads to the creation of seemingly random binary strings during the final phase, when all players apply the final $m$-fold Hadamard transform to their corresponding input registers. This means that if we assume that a tuple in the $|GHZ_{n+1}\rangle$ state is created instead of a tuple in the $|GHZ_n\rangle$ state, this $n + 1$-tuple will correspond to the $n$ players plus Eve. Accordingly, during the measurement phase, the results would be

$$|\psi_4\rangle = |\mathbf{a}\rangle_A\,|\mathbf{y}_{n-1}\rangle_E\,|\mathbf{y}_{n-2}\rangle_{n-2}\cdots|\mathbf{y}_0\rangle_0\,, \quad \text{for some} \quad \mathbf{a}, \mathbf{y}_0, \ldots, \mathbf{y}_{n-1} \in \{0,1\}^m\,, \quad (21)$$

instead of the anticipated

$$|\psi_4\rangle = |\mathbf{a}\rangle_A\,|\mathbf{y}_{n-2}\rangle_{n-2}\cdots|\mathbf{y}_0\rangle_0\,, \quad \text{for some} \quad \mathbf{a}, \mathbf{y}_0, \ldots, \mathbf{y}_{n-2} \in \{0,1\}^m\,. \quad (22)$$

In such a situation, Eve can be considered an extra player and, thus, her ability to acquire any extra information about the other players' measurement is, like all the other players, nonexistent.

### 6.4. Blinding Attack

Finally, we conclude our security analysis with the blinding attack. During this attack strategy, Eve, instead of trying to intercept the GHZ tuples, she blocks and destroys them entirely before they reach the intended players. Then she proceeds to create her own set of GHZ tuples, with a proper ancilla state in each tuple, and then distributes them to the players. From this description, it is obvious that in order for this particular type of attack to work, the entity responsible for the creation and distribution of the GHZ tuples must be a third party source and not a player. Therefore, during this attack, Eve will have a full set of tuples in the $|GHZ_{n+1}\rangle$ state, instead of the aforementioned smaller number of tuples in the $|GHZ_{n+1}\rangle$ state acquired, exploiting the inefficiency of our current photon sources during the PNS attack. However, once again, the scenario is similar to the PNS attack, meaning that Eve will be considered an extra player, and in that case, she will again be unable to acquire any information regarding the secret message.

## 7. Discussion and Conclusions

In this article, we introduced a new problem in the literature of cryptographic protocols, which we call the quantum secret aggregation problem. We gave a solution to the aforementioned problem that is based on the use of maximally entangled GHZ tuples. These are uniformly distributed among the players, which include the spymaster Alice and her network of agents, all of them being in different locations. We conducted a detailed analysis of the proposed protocol and, subsequently, illustrated its use with a toy scale example involving Alice and her two agents Bob and Charlie. Our presentation is completely general in the sense that the number of players can increase as needed, and the players are assumed to be spatially separated. It is clear that the same protocol can immediately accommodate groups of players that are in the same region of space.

In closing, we point out that the security of our protocol is attributed to its entanglement-based nature. For instance, entanglement monogamy precludes the entanglement of a maximally entangled tuple with any other qubit. This nullifies Eve's attempts at gaining

information by trying to entangle a qubit of the GHZ tuples used in our protocol during the transmission of the GHZ tuples to the players.

**Author Contributions:** Conceptualization, T.A. and M.A.; methodology, T.A.; validation, M.A.; formal analysis, T.A.; investigation, M.A.; writing—original draft preparation, M.A.; writing—review and editing, T.A.; visualization, M.A.; supervision, T.A.; project administration, T.A. and M.A. All authors have read and agreed to the published version of the manuscript.

**Funding:** This research received no external funding.

**Institutional Review Board Statement:** Not applicable.

**Informed Consent Statement:** Not applicable.

**Data Availability Statement:** Not Applicable, the study does not report any data.

**Conflicts of Interest:** The authors declare no conflict of interest.

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
