# Peer review of "Quantum Secret Aggregation Utilizing a Network of Agents"

_cryptography, doi:10.3390/cryptography7010005_

Round 1

Reviewer 1 Report

see enclosed comments

Reviewer 2 Report

The authors present a solution for a problem which called it Quantum Secret Aggregation. They proposed a method by using GHZ entanglement and solve it. I believe that it is amazing topic and it worth to publish it. However, I have one major consideration. I believe that although the work is interesting but the author has not emphasize the advantage and importance of it. I think the introduction parts needs to be improved. The authors should emphasize more about the importance and the applications of the work.  

Round 2

Reviewer 1 Report

see comments
